# Improved Physics-Informed Neural Networks Combined with Small Sample Learning to Solve Two-Dimensional Stefan Problem

**DOI:** 10.3390/e25040675

**Published:** 2023-04-18

**Authors:** Jiawei Li, Wei Wu, Xinlong Feng

**Affiliations:** College of Mathematics and System Sciences, Xinjiang University, Urumqi 830017, Chinawuwei837037@163.com (W.W.)

**Keywords:** Stefan problem, deep neural networks, small sample learning, efficient calculation method

## Abstract

With the remarkable development of deep learning in the field of science, deep neural networks provide a new way to solve the Stefan problem. In this paper, deep neural networks combined with small sample learning and a general deep learning framework are proposed to solve the two-dimensional Stefan problem. In the case of adding less sample data, the model can be modified and the prediction accuracy can be improved. In addition, by solving the forward and inverse problems of the two-dimensional single-phase Stefan problem, it is verified that the improved method can accurately predict the solutions of the partial differential equations of the moving boundary and the dynamic interface.

## 1. Introduction

For a fixed solution of a partial differential equation, it is usually restricted to a certain region. If this region varies with time, we call it the moving boundary problem. If part of the boundary of the fixed region is to be determined simultaneously with the solution of the fixed problem, we call it the free boundary problem, and the unknown boundary is called the free boundary. For free boundary problems, in addition to the usual fixed solution conditions, boundary conditions (Stefan conditions) must be added to the free boundary.

The Stefan problem is a class of heat conduction problems. It was first formulated by the Austrian physicist Joseph Stefan in the late 19th Century. The background of this problem is closely related to the industrial production during the industrial revolution of that time. In industrial production, many substances needed to be heat-treated, so the laws and methods of heat conduction needed to be studied to better control and utilize heat energy.

The difficulty of the Stefan problem is tracing the location of the interface, for example to model the process of change at the intersection of ice and water. Since the interface changes with time, these problems are also called free boundary problems. The study of free boundary problems has a wide practical background, such as plasma physics, percolation mechanics, and plasticity mechanics [1,2,3,4,5,6], which have presented various forms of constant and indeterminate free boundary problems. Furthermore, chemical vapor deposition [7], the vapor permeation of thermally cracked carbon in chemistry [8], tumor growth in medicine [9,10,11,12,13], the expansion propagation problem of biological populations [14,15,16,17], and the U.S. option pricing problem [18,19] also have free boundary problems. In fact, all free boundary problems are nonlinear problems, and it is important to solve them with the solution of the free boundary, which will be determined together with the solution of the fixed problem. Since these problems are closely related to practical applications, the efficient algorithmic implementation of free boundary problems is of great importance for scientific research and production practice.

Currently, a variety of numerical methods to the Stefan problems have emerged. The boundary integral method numerically solves the integral equation at a moving interface [20]. The interface tracing method explicitly represents the free boundary by using a fixed grid of points [1]. The immersion interface method [21] uses a fixed spatial grid for some physical quantities and a moving grid for the free boundary, and the information between the free boundary and the fixed grid is obtained by the immersion interface method. Segal [22] proposed an adaptive grid method for solving the free boundary problem, where the movement of the grid is determined by the control equations. Murray and Landis [23] compared the fixed-grid method with the adaptive-grid method. The adaptive-grid method can obtain the free boundary location more accurately, while the fixed-grid method can obtain certain physical quantities (temperature distribution over the whole solution area). More accurately, the enthalpy difference method [24] is an implicit method that represents the heat of the whole system by introducing an enthalpy function with a jump discontinuity at the free boundary, and this discontinuity helps to determine the location of the interface. The moving grid method [25] is a common method for solving free boundary problems, where the free boundary position is always fixed at the nth grid point, and the grid needs to be updated at each time step due to the movement of the free boundary. In recent years, the phase field method [26] has became a popular method based on the phase field function, which corresponds to a fixed constant in each phase and the interface region between the two values. This method considers a fuzzy boundary between the two phases, which is different from the classical Stefan problem, where the phase transition occurs in this interface region, where the thickness of the region is an artificially given parameter. The level-set approach [27] has also received increasing attention in recent years, by introducing a level-set function, which defines the interface position as a zero level set, obtained from the advection equation related to the velocity field, which varies considerably in different applications of the level-set approach. Sussman used the velocity of the fluid to model compressible two-phase flow [28], and Chen extended the interface moving velocity to the whole region by the advection equation in the solidification problem [27]. In contrast to the moving grid method, the level set method uses a fixed grid and avoids updating the grid at each time step. In contrast to the fixed-gridmethod, which finds the interface position at a fixed grid point, the phase-field method does not track the interface position precisely, and therefore, the discretization at the interface position is not as accurate as the fixed-grid method.

All the above methods have proven their high accuracy for some specific Stefan problems, and each has its own advantages and disadvantages. However, a general framework for solving the Stefan forward and inverse problems is still missing in all the methods at this stage.

In recent years, with the continuous development of neural networks, the use of neural networks to solve partial differential equations has gradually become popular [29,30,31,32]. Raissi et al. [33] proposed physics-informed neural networks for solving forward and inverse problems of partial differential equations, and deep-learning-based physics-informed neural networks have also been proposed for solving free boundary problems [34], as well as neural networks to solve the Stefan problems using a lattice-free grid-free automatic differentiation technique, which breaks the limitations of the above methods.

Deep learning models have achieved advanced results in solving various types of partial differential equations. However, the success of deep learning models relies heavily on a large amount of training data. In some specialized fields, the cost of data acquisition is very large. In addition, labeling samples requires much effort. Therefore, in recent years, a new learning approach, small sample learning, has gained popularity [35]. Small sample learning has been successfully applied to many new fields, such as: neural networks translation, target detection, etc. By using small sample learning, the accuracy of the model can be improved with few labeled data.

In this paper, we extended the recently emerging physics-informed neural networks framework [33] to solve the general Stefan problems. As we know, the original framework of physics-informed neural networks does not deal well with free boundary problems with time variation. To achieve this goal, we propose and modified the neural network framework proposed by Wang [34]. The specific contributions of this paper can be summarized as follows: Firstly, we incorporated the idea of small sample learning into neural network training, i.e., a small sample loss is added to the loss function optimization in order to improve the training accuracy and correct the model. Secondly, we changed the loss function to cope with the outliers that may arise from free boundary shifts. Finally, we applied the proposed framework to irregular regions and irregular free boundaries to test its performance. In summary, the proposed method provides a new general framework for solving the Stefan problems.

This paper is structured as follows: Section 2 introduces the Stefan problem and its mathematical model. Section 3 introduces the knowledge of neural networks and the PINNs’ improvement strategy. Section 4 verifies the accuracy and applicability of the proposed framework with numerical arithmetic examples. The conclusions and outlook are given in Section 5.

## 2. Model Issues

In this section, we introduce the mathematical model of the Stefan problem and the corresponding boundary conditions using a one-dimensional single-phase Stefan problem in the solidification or melting process as an example.

As shown in Figure 1, assume that a semi-infinite solid occupying 0⩽x<∞ is in the process of solidification or melting. For any moment t>0, the 0⩽x<∞ region consists of a solid and a liquid. The liquid is located in the $0⩽x<\alpha \left(t\right)$ region, and the solid is located in the $\alpha \left(t\right)<x<\infty$ region.

If the volume change due to solidification or melting is not considered and the region $0⩽x<\alpha \left(t\right)$ is considered, the temperature $u\left(x,t\right)$ satisfies the classical diffusion equation over the region:(1)$$\begin{array}{c} \frac{\partial u\left(x,t\right)}{\partial t}=\frac{\partial^{2}u\left(x,t\right)}{\partial x^{2}},\;\;x\in \left(0,\alpha \left(t\right)\right),\;\;t\in \left(0,T\right), \end{array}$$
and for the initial and boundary conditions:(2)$$\begin{array}{cc} u\left(x,0\right)= & \;\;u_{0}\left(x\right),\;\;x\in \left[0,\alpha \left(0\right)\right], \end{array}$$
(3)$$\begin{array}{cc} u\left(0,t\right)= & \;\;h\left(t\right),\;\;\;\;\;\;\;t\in \left[0,T\right]. \end{array}$$
at the interface $\alpha \left(t\right)$, the following Stefan conditions need to be met: (4)$$\begin{array}{cc} \alpha \left(0\right)= & \;\;\alpha_{0}, \end{array}$$
(5)$$\begin{array}{cc} u\left(\alpha \left(t\right),t\right)= & \;\;0,\;\;\;\;\;t\in \left[0,T\right], \end{array}$$
(6)$$\begin{array}{cc} \frac{\partial u}{\partial x}\left(\alpha \left(t\right),t\right)= & \;\;g\left(t\right),\;\;\;t\in \left[0,T\right], \end{array}$$
where $\alpha \left(t\right)$ denotes the free boundary, (4) denotes the initial position of the free boundary, and (5) denotes the temperature at the time of freezing. For the forward problem, in thermal physics, it is the simultaneous solution of the temperature distribution and the free boundary for which various parameters are known. Each Stefan problem corresponds to a class of inverse problems, and similar to other mathematical physics inverse problems, the inverse Stefan problem is not definite, while the uniqueness and stability of the solution are not always guaranteed.

## 3. Numerical Methods

In this section, the main methods and ideas of feedforward neural networks and physics-informed neural networks for solving partial differential equations are reviewed, and a general model of improved physics-informed neural networks combined with small sample learning for solving the Stefan problem is proposed.

### 3.1. Feedforward Neural Networks

Mathematically, a neural network is a specific class of complex functions, the simplest of which is a feedforward neural network (FNN), which is also called a multilayer perceptron (MLP). Let $F^{P}\left(x\right):R^{d_{in}}\to R^{d_{out}}$ be a *P*-layer feedforward neural networks with P−1 hidden layers; the number of neurons in the input layer is $N_{0}=d_{in}$, and the number of neurons in the output layer is $N_{P}=d_{out}$. The network weight of the *p*th layer is $W^{p}\in R^{N_{p}\times N_{p}-1}$, and the network bias is $b^{p}\in R^{N_{p}}$. If given the activation function σ, the feedforward neural network (FNN) can be expressed as
(7)$$\begin{array}{cc} \; & F^{0}x=x\in R^{d_{\mathrm{in}}},\\ \; & F^{p}x=\sigma \left(W^{p}F^{p-1}\left(x\right)+b^{p}\right)\in R^{N_{p}},\\ \; & F^{P}x=W^{P}F^{P-1}\left(x\right)+b^{P}\in R^{d_{\mathrm{out}}}. \end{array}$$

### 3.2. Physics-Informed Neural Networks

We considered the following parameterized partial differential equation:(8)$$\begin{array}{c} F\left(x,t;\partial_{x}u,\partial_{t}u,\ldots ;\lambda \right)=0,\;\;\left(x,t\right)\;\in \;\Omega \times \left(0,T\right],\\ u\left(x,0\right)=g\left(x\right),\;\;\;x\;\in \Omega ,\\ Bu\left(x,t\right)=h\left(x,t\right),\;\;\left(x,t\right)\in \;\;\partial \Omega \times \left[0,T\right]. \end{array}$$
where $x\in \Omega \in R^{n}$, *F* denotes the residuals of the partial differential equation, *F* in $\left(\partial_{x}u,\partial_{t}u,\ldots \right)$ denotes the space–time differential operator, $\lambda =\left[\lambda_{1},\lambda_{2},\lambda_{3}\ldots \right]$ represents the parameters of the partial differential equation, $u\left(x,t\right)$ is the solution of the partial differential equation, Ω denotes the solution region of the partial differential equation, ∂Ω is the boundary of Ω, and B represents any of the boundary operators, including Dirichlet, Neumann, Robin, and periodic.

PINNs [33] use a fully connected feedforward neural networks consisting of multiple hidden layers to approximate the solution of the partial differential equation with spatio-temporal coordinates $u\left(x,t\right)$ as the input. The neural network is displayed in Figure 2.

First, we establish the loss function based on the form of the partial differential equation and the initial margin value condition: (9)$$\begin{array}{c} L_{\mathrm{total}}=\lambda_{f}L_{f}+\lambda_{\mathrm{ic}}L_{\mathrm{ic}}+\lambda_{\mathrm{bc}}L_{\mathrm{bc}}, \end{array}$$
where
(10)$$\begin{array}{cc} \; & L_{f}=\frac{1}{N_{f}}\sum_{i=1}^{N_{f}}{∥F\left(x,t;\partial_{x}u,\partial_{t}u,\ldots ;\lambda \right)∥}_{2}^{2},\\ \; & L_{\mathrm{ic}}=\frac{1}{N_{\mathrm{ic}}}\sum_{j=1}^{N_{\mathrm{ic}}}{∥u\left(x_{\mathrm{ic}}^{j},0\right)-g\left(x_{\mathrm{ic}}^{j}\right)∥}_{2}^{2},\\ \; & L_{\mathrm{bc}}=\frac{1}{N_{\mathrm{bc}}}\sum_{k=1}^{N_{\mathrm{bc}}}{∥B\left(u\left(x_{\mathrm{bc}}^{k},t_{\mathrm{bc}}^{k}\right)\right)-h\left(x_{\mathrm{bc}}^{k},t_{\mathrm{bc}}^{k}\right)∥}_{2}^{2}, \end{array}$$
where $L_{f}$, $L_{\mathrm{ic}}$, and $L_{\mathrm{bc}}$ denote the loss functions satisfying the control equations, initial conditions, and boundary conditions and $\lambda_{f}$, $\lambda_{\mathrm{ic}}$, and $\lambda_{\mathrm{bc}}$ denote the weights of the loss functions $L_{f}$, $L_{\mathrm{ic}}$, and $L_{\mathrm{bc}}$, respectively, where the derivatives involved are automatically differentiated by the neural networks.

After the loss function $L_{\mathrm{total}}$ is established, the optimal parameters $\theta^{*}=\left(W^{*},b^{*}\right)$ of the neural networks are obtained by iterative updating with the objective of minimizing the loss function, considering that the loss function is nonlinear and nonconvex for $\theta^{*}$, so a gradient-based optimizer is used to minimize the loss function, such as: Adagrad, AdaDelta, Adam, momentum, and RMSProp.

### 3.3. Neural Network Improvement Strategies for Stefan Problem

PINNs have been applied in various scientific and engineering fields to solve various complex physical problems. In the free boundary problem, the simple PINNs can no longer meet the requirements of the Stefan problem, and certain structures need to be changed accordingly. Therefore, we propose the following improvement strategies.

#### 3.3.1. Neural Networks’ Basic Structure

Recall that, for the Stefan problem, we aim to find both the temperature solution $u\left(x,t\right)$ and the moving boundary $\alpha \left(t\right)$. To this end, we build two deep neural networks $u_{\theta }\left(x,t\right)$ and $\alpha_{\delta }\left(t\right)$ with θ and δ as independent parameter spaces to approximate the solutions $u\left(x,t\right)$ and $\alpha \left(t\right)$. Then, we approximate the area temperature solution and the moving boundary by minimizing a composite loss function.

#### 3.3.2. Adding Additional Terms to the Loss Function

The original PINN loss function is shown in (9), and in this paper, we considered adding a loss function with small sample data, as follows:(11)$$\begin{array}{c} L_{\mathrm{total}}=\lambda_{f}L_{f}+\lambda_{\mathrm{ic}}L_{\mathrm{ic}}+\lambda_{\mathrm{bc}}L_{\mathrm{bc}}+\lambda_{\mathrm{ssl}}L_{\mathrm{ssl}}, \end{array}$$
where $L_{ssl}$ is the loss function for the small sample data, which is used to correct the model. Adding small sample data can improve the prediction accuracy of the model, especially in the case of fewer data. Meanwhile, it can improve the generalization ability and robustness of the model to avoid the risk of overfitting.

#### 3.3.3. Loss Function Improvement Strategy

The mean-squared error (MSE) is the most-commonly used regression loss function in deep learning and is the sum of squares of the distance between the target variable and the predicted value. Considering the specificity of Stefan problem, sharp or irregular regions will appear during the melting or freezing process, which leads to outliers, and the mean-squared error is very sensitive to outliers; thus, we introduce the log-cosh loss function, for a small error; $\log\left(\cosh\left(x\right)\right)$ is similar to $\frac{x^{2}}{2}$, while for large errors, $\log\left(\cosh\left(x\right)\right)$ is similar to $\left|x\right|-\log2$, which means that the log-cosh loss function can have the advantage of the mean-squared error while not being affected by too many outliers. It is also quadratically derivable at every point. The introduction of the log-cosh loss function allows for better model generality and model prediction accuracy. The log-cosh loss function is expressed as follows:(12)$$\begin{array}{c} L\left(Y,\hat{Y}\right)=\sum_{i=1}^{n}\log\left(\cosh\left({\hat{Y}}_{i}-Y_{i}\right)\right), \end{array}$$
where the log-cosh loss function is smooth and has a continuous derivative property, which allows the model to be trained using derivative-based optimization algorithms such as gradient descent, and the log-cosh loss function is a symmetric loss function; it has equal loss for positive and negative errors. This symmetry allows the model to better handle both positive and negative errors, thus improving the model fitting ability.

To summarize, our proposed algorithm is displayed in Algorithm 1 and visualized in the schematic diagram in Figure 3, where we explain each step in detail. First, we construct two neural networks $u_{\theta }\left(x,t\right)$ and $\alpha_{\delta }\left(t\right)$. $\theta ={\left\{W^{k},b^{k}\right\}}_{1⩽k⩽K}$ is the set of weight matrices and bias vectors in u; $\delta ={\left\{W^{j},b^{j}\right\}}_{1⩽j⩽J}$ is the set of weight matrices and bias vectors in α; for the neural networks as a substitute for *u* and α, we can use auto-differentiation and the chain rule to differentiate the functions, then we need to restrict the two neural networks to satisfy the physical conditions imposed by the PDE and Stefan condition, while it is harder to achieve if we constrain over the whole region, so we constrain *u* and α at scattered points and various training sets and measure the difference between the two neural networks and the constraint; we define the loss function as the sum of the hyperbolic cosine logarithms of the equations and the residuals of the boundary conditions (where the derivatives involved are handled automatically by AD), and in the last step, we minimize the total loss function to obtain the optimal parameters and, thus, the optimal solution. Since the loss is nonlinear and nonconvex, we usually use gradient-based optimizers to minimize the loss function, such as gradient descent, Adam [36], and L-BFGS [37].
**Algorithm 1** Deep neural networks for solving the Stefan problem.1:Two neural networks $u_{\theta }\left(x,t\right)$ and $\alpha_{\delta }\left(t\right)$ with θ and δ as parameters and propagating forward independently of each other are constructed.2:The training sets $K_{f}$ for the temperature control equation, the training sets $K_{u_{0}}$ and $K_{u_{\mathrm{nc}}}$ for the initial margin condition, the training sets $K_{\delta_{0}}$ for the initial margin condition of the moving interface, $K_{\delta_{\mathrm{nc}}}$, $K_{\delta_{\mathrm{ic}}}$, $K_{\delta_{\mathrm{bc}}}$, and the training set $K_{u_{\mathrm{ssl}}}$ for the residuals of the internal small sample data.3:The total loss function is assigned by summing the hyperbolic cosine logarithms of the control equation, the initial margin conditional residuals, the moving boundary initial margin conditional residuals, and the interior point residuals.4:The optimal parameters θ and δ are obtained by minimizing the loss function and training the neural network to obtain the optimal solution.

## 4. Numerical Examples

In this section, we consider several two-dimensional classical forward and inverse Stefan problems to demonstrate the robustness and power of the proposed model. We validated the proposed model for different numbers of small sample data points (0, 50, 100, 200). The results showed that small sample data can effectively improve the detection accuracy. In Section 4.1, one two-dimensional forward Stefan problem is given to test the accuracy of the proposed model, and in Section 4.2, two two-dimensional inverse Stefan problems are given to verify the generality of the model.

For error evaluation, we used the relative $L^{2}$ norm: (13)$$\begin{array}{c} Error=\frac{\sqrt{\sum_{j=1}^{M}|u_{\mathrm{exact}}^{j}-u_{\mathrm{pred}}^{j}{|}^{2}}}{\sqrt{\sum_{j=1}^{M}|u_{\mathrm{exact}}^{i}{|}^{2}}}. \end{array}$$
Here, M represents the number of all points in the training process of two mutually independent neural networks, $u_{\mathrm{pred}}$ represents the predicted values of the corresponding coordinate points, and $u_{\mathrm{exact}}$ represents the exact values of the corresponding coordinate points.

### 4.1. Two-Dimensional Single-Phase Stefan Forward Problem

In this subsection, a classical two-dimensional solid melting problem model is used as an example to illustrate the effectiveness of the method [38].

As depicted in Figure 4, let $D\in R^{2}$, for any 0<t<T, $\Omega \left(t\right)$, be the time-dependent bounded subdomain in *D*, $\partial \Omega \left(t\right)$ be its boundary, and $\partial \Omega \left(t\right)=\Gamma \left(t\right)\cup \Sigma \left(t\right)$, where Γ= $\cup_{0<t<T}\Gamma \left(t\right)$ represents the fixed boundary and $\Sigma =\cup_{0<t<T}\Sigma \left(t\right)$ represents the moving boundary.

The above Stefan problem can be described in mathematics as
(14)$$\begin{array}{cc} \; & u_{t}-\Delta u=0\;\;in\Omega , \end{array}$$
(15)$$\begin{array}{cc} \; & {u|}_{\bar{\Gamma }}=g, \end{array}$$
(16)$$\begin{array}{cc} \; & {u|}_{\Omega \left(0\right)}=u_{0}, \end{array}$$
(17)$$\begin{array}{cc} \; & \Sigma \left(0\right)=\Sigma_{0}, \end{array}$$
(18)$$\begin{array}{cc} \; & {u|}_{\bar{\Sigma }}=u^{*}, \end{array}$$
(19)$$\begin{array}{cc} \; & \frac{\partial u}{\partial n}{|}_{\bar{\Sigma }}=h, \end{array}$$
where Δ is the Laplace operator for spatial variables, *n* is the normal vector of the definition field Ω with respect to the time variables, and the definition field Ω is
(20)$$\begin{array}{c} \Omega \left(t\right)=\left\{\left(x,y,t\right)\in R^{3}:0<x<\alpha \left(y,t\right),0<y<1\right\}\subset \Omega^{*}. \end{array}$$

$\alpha \left(y,t\right)$ denotes the unknown free boundary; the artificial region is defined as: $\Omega^{*}=$$\left[0,2.25\right]\times \left[0,1\right]\times \left[0,1\right]$, defining $\Sigma \left(t\right)$ for
(21)$$\begin{array}{c} \Sigma \left(t\right)=\left\{\left(x,y,t\right)\in R^{3}:\Phi \left(x,y,t\right)=0,0<y<1,0⩽t⩽1\right\}, \end{array}$$
where $\Phi \left(x,y,t\right)=x-\alpha \left(y,t\right)$. Define the following parameters:(22)$$\begin{array}{cc} \; & u_{0}\left(x,y\right)=\exp\left(-x+\frac{1}{2}y+\frac{1}{2}\right),\;\;\;\;x,y\in \Omega \left(0\right),\\ \; & u\left(\alpha \left(y,t\right),y,t\right)=u^{*}=0,\\ \; & h=\frac{1}{\left|\nabla \Phi \right|}\frac{\partial \Phi }{\partial t},\;\;\alpha \left(y,0\right)=\alpha_{0}\left(y\right)=\frac{1}{2}y+\frac{1}{2}. \end{array}$$

The boundary conditions are defined as
(23)$$\begin{array}{cc} \; & u\left(x,0,t\right)=g_{1}\left(x,t\right)=\exp\left(1.25t-x+\frac{1}{2}\right)-1,\;\;\left(x,0,t\right)\in \Omega ,\\ \; & u\left(0,y,t\right)=g_{2}\left(x,t\right)=\left(1.25t+0.5y+\frac{1}{2}\right)-1,\;\;\left(0,y,t\right)\in \Omega ,\\ \; & u\left(x,1,t\right)=g_{3}\left(x,t\right)=\exp\left(1.25t-x+1\right),\;\;\left(x,1,t\right)\in \Omega . \end{array}$$

Construct the following true solution:(24)$$\begin{array}{cc} \; & u\left(x,y,t\right)=\exp\left(\frac{5}{4}t-x+\frac{1}{2}y+\frac{1}{2}\right)-1,\\ \; & \alpha \left(y,t\right)=\frac{1}{2}y+\frac{5}{4}t+\frac{1}{2}. \end{array}$$

As shown in Algorithm 1, we built two fully connected neural networks $u_{\theta }\left(x,y,t\right)$ and $\alpha_{\delta }\left(y,t\right)$ that propagate independently of each other to approximate the solution of the control equation $u\left(x,y,t\right)$ and the moving boundary $\alpha \left(y,t\right)$, and we trained these two neural networks by minimizing the loss function.
(25)$$\begin{array}{c} L\left(\theta ,\delta \right)=L_{f}\left(\theta \right)+L_{u_{0}}\left(\theta \right)+L_{u_{\mathrm{bc}}}\left(\theta \right)+L_{\alpha_{0}}\left(\delta \right)+L_{\alpha_{\mathrm{bc}}}\left(\theta ,\delta \right)+L_{\alpha_{\mathrm{nc}}}\left(\theta ,\delta \right)+L_{u_{\mathrm{ssl}}}\left(\theta \right), \end{array}$$
where
$$\begin{array}{c} L_{f}\left(\theta \right)=\sum_{i=1}^{N_{f}}\log\left(\cosh\left(u_{\theta }\left(x_{f}^{i},y_{f}^{i},t_{f}^{i}\right)-u\left(x_{f}^{i},y_{f}^{i},t_{f}^{i}\right)\right)\right), \end{array}$$
$$\begin{array}{c} L_{u_{0}}\left(\theta \right)=\sum_{j=1}^{N_{u_{0}}}\log\left(\cosh\left(u_{\theta }\left(x_{u_{0}}^{j},y_{u_{0}}^{j},0\right)-u_{0}\left(x_{u_{0}}^{j},y_{u_{0}}^{j},0\right)\right)\right), \end{array}$$
$$\begin{array}{c} L_{u_{\mathrm{bc}}}\left(\theta \right)=\sum_{k=1}^{N_{u_{\mathrm{bc}}}}\log\left(\cosh\left(u_{\theta }\left(x_{u_{\mathrm{bc}}}^{k},0,t_{u_{\mathrm{bc}}}^{k}\right)-g_{1}\left(x_{u_{\mathrm{bc}}}^{k},t_{u_{\mathrm{bc}}}^{k}\right)\right)\right)\\ +\sum_{k=1}^{N_{u_{\mathrm{bc}}}}\log\left(\cosh\left(u_{\theta }\left(0,y_{u_{\mathrm{bc}}}^{k},t_{u_{\mathrm{bc}}}^{k}\right)-g_{2}\left(y_{u_{\mathrm{bc}}}^{k},t_{u_{\mathrm{bc}}}^{k}\right)\right)\right)\\ +\sum_{k=1}^{N_{u_{\mathrm{bc}}}}\log\left(\cosh\left(u_{\theta }\left(x_{u_{\mathrm{bc}}}^{k},1,t_{u_{\mathrm{bc}}}^{k}\right)-g_{2}\left(x_{u_{\mathrm{bc}}}^{k},t_{u_{\mathrm{bc}}}^{k}\right)\right)\right), \end{array}$$
$$\begin{array}{c} L_{\alpha_{0}}\left(\delta \right)=\sum_{j=1}^{N_{\alpha_{0}}}\log\left(\cosh\left(\alpha_{\delta }\left(y_{\alpha_{0}}^{j},0\right)-\alpha_{0}\left(y_{\alpha_{0}}^{j}\right)\right)\right), \end{array}$$
$$\begin{array}{c} L_{\alpha_{\mathrm{bc}}}\left(\theta ,\delta \right)=\sum_{k=1}^{N_{\alpha_{\mathrm{bc}}}}\log\left(\cosh\left(u_{\theta }\left(\alpha_{\delta }\left(y_{{}_{\mathrm{bc}}}^{k},t_{{}_{\mathrm{bc}}}^{k}\right),y_{{}_{\mathrm{bc}}}^{k},t_{{}_{\mathrm{bc}}}^{k}\right)\right)\right), \end{array}$$
$$\begin{array}{c} L_{\alpha_{\mathrm{nc}}}\left(\theta ,\delta \right)=\sum_{k=1}^{N_{\alpha_{\mathrm{nc}}}}\log\left(\cosh\left(\frac{\partial u_{\theta }}{\partial n}\left(\alpha_{\delta }\left(y_{{}_{\mathrm{nc}}}^{k},t_{{}_{\mathrm{nc}}}^{k}\right),y_{{}_{\mathrm{nc}}}^{k},t_{{}_{\mathrm{nc}}}^{k}\right)-h\left(y_{{}_{\mathrm{nc}}}^{k},t_{{}_{\mathrm{nc}}}^{k}\right)\right)\right), \end{array}$$
$$\begin{array}{c} L_{u_{\mathrm{ssl}}}\left(\theta \right)=\sum_{j=1}^{N_{u_{\mathrm{ssl}}}}\log\left(\cosh\left(u_{\theta }\left(x_{{}_{\mathrm{ssl}}}^{j},y_{{}_{\mathrm{ssl}}}^{j},t_{{}_{\mathrm{ssl}}}^{j}\right)-u\left(x_{{}_{\mathrm{ssl}}}^{j},y_{{}_{\mathrm{ssl}}}^{j},t_{{}_{\mathrm{ssl}}}^{j}\right)\right)\right). \end{array}$$
In particular, since the moving interface $\alpha \left(y,t\right)$ is unknown in advance, the residual point region of the equation was set to the artificial region $\Omega^{*}$, and the neural networks was trained to restrict the neural network prediction solution to the defined domain Ω. Two Stefan neural networks $u_{\theta }\left(x,t\right)$ and $\alpha_{\delta }\left(t\right)$ were constructed with independent forward propagation, and the neural network parameters were set as follows: $u_{\theta }\left(x,t\right)$ has a network structure of 3+100×3+1; the input layer has three variables and three hidden layers; each layer contains 100 neurons; the output layer outputs the neural networks prediction solution. The network structure of $\alpha_{\delta }\left(t\right)$ is 2+100×3+1; similarly, the input layer has two variables, and the output layer has three hidden layers containing 100 neurons each, while the moving boundary of the neural network prediction is the output. The training set was selected as $N_{f}$ with 256 points, 256 points for each of the initial border conditions (including the initial border conditions on the free boundary), and 0, 50, 100, and 200 for the small sample data points in the region. The results are shown in Table 1. The same initial learning rate of $10^{-3}$ was set for both neural networks, and the number of iterations was 40,000 with Adam’s algorithm, using tanh as the activation function and using Xavier to initialize both neural networks.

By training the neural networks with the above parameters, Figure 5 shows the $L^{2}$ error plot of the predicted solution and the true solution, and Figure 6 shows the $L^{2}$ error plot of the predicted free interface $\alpha \left(y,t\right)$ and the true free boundary of the neural networks. It is worth mentioning that the number of small samples $u_{ssl}$ is set to 100 in Figure 5 and Figure 6, and we can see that the $L^{2}$ errors reached $8.57\times 10^{-3}$ and $3.25\times 10^{-2}$, respectively, which is a significant improvement in accuracy compared with the original neural networks [34].

### 4.2. Two-Dimensional Single-Phase Stefan Inverse Problem I

For the two-dimensional single-phase Stefan inverse problem, we assumed that the boundary conditions are not known and provide additional information at the final moment T:(26)$$\begin{array}{c} {\left.u\right|}_{\Omega \left(T\right)}=u_{T}. \end{array}$$

Then, this class of inverse problems means that we know the data at the moment T = 1 and need to find the temperature solution $u\left(x,y,t\right)$ and the free boundary $\alpha \left(y,t\right)$ and satisfy (14), (16), and (17)–(19).
(27)$$\begin{array}{c} u\left(x,y,T\right)=u_{T}=\exp\left(1.25-x+0.5y+0.5\right)-1. \end{array}$$

As shown in Algorithm 1, we built two fully connected neural networks $u_{\theta }\left(x,y,t\right)$ and $\alpha_{\delta }\left(y,t\right)$ that propagate independently of each other to approximate the solution of the control equation $u\left(x,y,t\right)$ and the moving boundary $\alpha \left(y,t\right)$, and we trained these two neural networks by minimizing the loss function.
(28)$$\begin{array}{c} L\left(\theta ,\delta \right)=L_{r}\left(\theta \right)+L_{u_{0}}\left(\theta \right)+L_{u_{T}}\left(\theta \right)+L_{\alpha_{\mathrm{bc}}}\left(\theta ,\delta \right)+L_{\alpha_{\mathrm{nc}}}\left(\theta ,\delta \right)+L_{\alpha_{0}}\left(\delta \right)+L_{u_{\mathrm{ssl}}}\left(\theta \right), \end{array}$$
where the loss function definitions are all the same as in Section 4.1, except that $L_{u_{T}}\left(\theta \right)$:(29)$$\begin{array}{c} L_{u_{T}}\left(\theta \right)=\sum_{j=1}^{m}\log\left(\cosh\left(u_{\theta }\left(x^{j},y^{j},1\right)-u\left(x^{j},y^{j},1\right)\right)\right). \end{array}$$

We trained the neural networks using exactly the same parameters as in Section 4.1, as shown in Table 2, which visualizes the $L^{2}$ error of the solution and moving boundaries when the small sample data are 0, 50, 100, 150, and 200, respectively.

Figure 7 gives the error plots of the predicted and exact temperature solutions at moments *t* = 0.2, 0.4, 0.6, and 0.8, and Figure 8 shows the error plots of the predicted moving boundaries and the true moving boundaries. In Figure 7 and Figure 8, the number of small samples $u_{ssl}$ was set to 100, and we can see that the $L^{2}$ error reached $3.66\times 10^{-2}$ and $3.15\times 10^{-2}$, which improved the accuracy of the neural networks compared with the original neural networks.

### 4.3. Two-Dimensional Single-Phase Stefan Inverse Problem II

This part mainly discusses the Stefan problem of inversion of the phase change boundary in the case of known temperature and heat changes at the boundary of a homogeneous medium, without considering any initial conditions or boundary conditions at the moving interface; only some given temperature measurements in the definition domain were considered, and neural networks were used to approximate the temperature solution and the unknown position of the moving boundary, then the above inverse problem is described by the mathematical formula:(30)$$\begin{array}{cc} \; & u_{t}-\Delta u=0,\;\left(x,y\right)\in \Omega ,\;\;\\ \; & \Sigma \left(0\right)=\Sigma_{0},\\ \; & {u|}_{\bar{\Sigma }}=u^{*},\;\;\\ \; & \frac{\partial u}{\partial n}{|}_{\bar{\Sigma }}=h. \end{array}$$

Similarly, two fully connected neural networks $u_{\theta }\left(x,y,t\right)$ and $\alpha_{\delta }\left(y,t\right)$ that propagate independently of each other were constructed to approximate the solution of the equation $u\left(x,y,t\right)$ and the moving boundary $\alpha \left(y,t\right)$, and we trained these two neural networks by minimizing the following loss functions:(31)$$\begin{array}{c} L\left(\theta ,\delta \right)=L_{\mathrm{data}}\left(\theta \right)+L_{f}\left(\theta \right)+L_{\alpha_{\mathrm{bc}}}\left(\theta ,\delta \right)+L_{\alpha_{\mathrm{nc}}}\left(\theta ,\delta \right)+L_{u_{\mathrm{ssl}}}\left(\theta \right), \end{array}$$
where
$$\begin{array}{c} L_{data}\left(\theta \right)=\sum_{i=1}^{N_{data}}\log\left(\cosh\left(u_{\theta }\left(x_{data}^{i},y_{data}^{i},t_{data}^{i}\right)-u^{i}\left(x^{i},y^{i},t^{i}\right)\right)\right), \end{array}$$
$$\begin{array}{c} L_{f}\left(\theta \right)=\sum_{i=1}^{N_{f}}\log\left(\cosh\left(u_{\theta }\left(x_{f}^{i},y_{f}^{i},t_{f}^{i}\right)-u\left(x_{f}^{i},y_{f}^{i},t_{f}^{i}\right)\right)\right), \end{array}$$
$$\begin{array}{c} L_{\alpha_{\mathrm{bc}}}\left(\theta ,\delta \right)=\sum_{k=1}^{N_{\alpha_{\mathrm{bc}}}}\log\left(\cosh\left(u_{\theta }\left(\alpha_{\delta }\left(y_{\mathrm{bc}}^{k},t_{\mathrm{bc}}^{k}\right),y_{\mathrm{bc}}^{k},t_{\mathrm{bc}}^{k}\right)\right)\right), \end{array}$$
$$\begin{array}{c} L_{\alpha_{\mathrm{nc}}}\left(\theta ,\delta \right)=\sum_{k=1}^{N_{\alpha_{\mathrm{nc}}}}\log\left(\cosh\left(\frac{\partial u_{\theta }}{\partial n}\left(\alpha_{\delta }\left(y_{\mathrm{nc}}^{k},t_{nc}^{k}\right),y_{\mathrm{nc}}^{k},t_{\mathrm{nc}}^{k}\right)-h\left(y_{\mathrm{nc}}^{k},t_{\mathrm{nc}}^{k}\right)\right)\right), \end{array}$$
$$\begin{array}{c} L_{u_{\mathrm{ssl}}}\left(\theta \right)=\sum_{j=1}^{N_{u_{\mathrm{ssl}}}}\log\left(\cosh\left(u_{\theta }\left(x_{\mathrm{ssl}}^{j},y_{\mathrm{ssl}}^{j},t_{\mathrm{ssl}}^{j}\right)-u\left(x_{\mathrm{ssl}}^{j},y_{\mathrm{ssl}}^{j},t_{\mathrm{ssl}}^{j}\right)\right)\right). \end{array}$$

In particular, the first loss function was constructed from the residuals of the temperature observation points in the region, where ${\left\{\left(x^{i},y^{i},t^{i}\right),u^{i}\right\}}_{i=1}^{N}$ is a randomly sampled data point in the domain. The neural networks parameters were set as follows: the network structures of the two neural networks $u_{\theta }\left(x,t\right)$ and $\alpha_{\delta }\left(t\right)$ were the same as in the previous example, and the training set was selected as follows: the residuals of the equations and the initial margin conditions (including the initial margin conditions on the free boundary) were each selected as 256 points. The results are shown in Table 3. The initial learning rate was set to $10^{-3}$ for both neural networks, and the number of iterations was set to 40,000 and optimized by Adam optimizer. Xavier initialized both neural networks.

From Figure 9 and Figure 10, it can be seen that the $L^{2}$ error of the solution of the inverse problem and the moving interface can reach $9.97\times 10^{-3}$ and $1.26\times 10^{-2}$ for the improved deep neural networks with small sample data of 100, which is a significant improvement compared with the original neural network.

The above two-dimensional single-phase Stefan inverse problem II is based on the temperature measurement value M=50, and then, we tested the performance of our proposed neural network framework by adding white noise κ with magnitude equal to 1%, 2%, 5%, and 10% of the $L^{\infty }$ norm of the solution function $u\left(x,y,t\right)$, where the small sample data points were 100. The relative $L^{2}$ errors obtained for the prediction solutions $u\left(x,y,t\right)$ and $\alpha \left(y,t\right)$ are shown in Table 4 and Table 5, respectively. We can observe that the prediction accuracy of both $u\left(x,y,t\right)$ and $\alpha \left(y,t\right)$ improved as the total number of temperature measurement data *M* increased, but became lower as the noise level κ increased. The latter case verified our conjecture that the dataset became increasingly inaccurate due to higher levels of noise. Another important observation is that, given a sufficient amount of temperature measurement data (e.g., M=200), even with noise corruption up to 10%, the relative $L^{2}$ error between the predicted solutions $u\left(x,y,t\right)$ and $\alpha \left(y,t\right)$ can reach $6.78\times 10^{-2}$ and $3.77\times 10^{-2}$, and it can be seen that the improved physics-informed neural networks with small sample learning can accurately identify the moving boundaries despite the presence of a large amount of noisy data.

Figure 11 shows the drop plots of the loss function for the three cases when the noise level κ is 10%, and it can be seen that the loss function image became relatively stable with the addition of the small sample loss function, which also validated the relevant narratives in Section 3.3.2 and Section 3.3.3.

### 4.4. Irregular Area Stefan Problem

To make our proposed algorithmic framework more convincing, the main work of this subsection is to consider numerical solutions and the free boundary of the two-dimensional Stefan problem on an L-shaped complex computational domain that differs from the traditional $\left[a,b\right]\times \left[c,d\right]$ rectangular domain, which is more capable of testing the performance of our algorithmic framework. It is worth mentioning that the equations and Stefan conditions used in this section are the same as in Section 4.1, except that the boundary information is different due to the complex computational domain.

Figure 12 represents the plan view of the selected training points on the L-shaped complex computational domain. Unlike the above example, we randomly selected the training points on the boundary, and after the complex computational domain was triangulated and dissected, the internal points were selected at the nodes instead of selecting random samples as internal information.

We used the same hyperparameters in the numerical example above and constructed the same loss function in (15), minimizing the loss function by the Adam optimization algorithm to obtain the optimal solution and the free boundary, as shown in Figure 13, where we can see that our proposed method can also achieve good accuracy for complex computational domains. The relative $L^{2}$ error between the predicted solutions $u\left(x,y,t\right)$ and $\alpha \left(y,t\right)$ can reach $5.74\times 10^{-3}$ and $1.56\times 10^{-2}$. Snapshots of moments for t=0.2 s, t=0.4 s, and t=0.8 s are given. It can be seen that the model predicts the temperature solution and moving boundary, in good agreement with the exact solution.

### 4.5. Irregular Free Boundary Stefan Problem

In this subsection, consider an irregular free boundary problem, the initial shape of which is described by a unit circle of $x^{2}+y^{2}=1$ with equations and boundary conditions as described in Section 4.1, $t\in \left[0,1\right]$, and consider the following analytic solution:(32)$$\begin{array}{c} u\left(x,y,t\right)=e^{t}\left(\left(t+1\right)x^{2}+\left(5t+1\right)y^{2}-t^{2}-1\right) \end{array}$$
(33)$$\begin{array}{c} \alpha \left(x,y,t\right)=\left(t+1\right)x^{2}+\left(5t+1\right)y^{2}-t^{2}-1 \end{array}$$

Similar to the description in the previous section, we randomly selected training points on the boundary, and after the complex computational domain was triangulated and dissected, internal points were selected at the nodes. The distribution of training points is shown in the Figure 14.

Next, we also used the same loss function construction as in (15) to predict $u\left(x,y,t\right)$ and $\alpha \left(x,y,t\right)$. We give snapshots of the predictions when t=0.2 s, t=0.4 s, and t=0.8 s in Figure 15. From that, we can see that our proposed neural network framework can also have a good prediction for general irregular moving boundaries; meanwhile, the relative $L^{2}$ error between the predicted solutions $u\left(x,y,t\right)$ and $\alpha \left(x,y,t\right)$ can reach $3.51\times 10^{-2}$ and $2.47\times 10^{-2}$.

## 5. Conclusions

In this paper, by improving the neural network structure of physics-informed neural networks (PINNs) and combining the idea of small sample learning, a generalized neural network framework for solving the Stefan problems was proposed, which can be directly applied to various types of Stefan problems with only small changes. After several numerical experiments, it was proven that the improved deep neural networks based on small sample learning improved the computational accuracy by 2–3-times compared with the original neural network, and the model generalization ability was significantly improved. Meanwhile, this paper demonstrated that the proposed method had a good prediction effect and accuracy through the examples of the irregular region and irregular free boundary. The study of Stefan problems with sharp, irregular geometries and topological variations, as well as three-dimensional or even higher-dimensional Stefan problems is the focus of the future work.

## Figures and Tables

**Figure 1 entropy-25-00675-f001:**
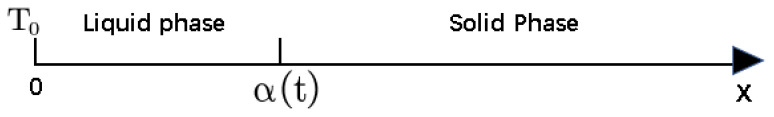
One-dimensional single-phase Stefan problem solidification or melting model.

**Figure 2 entropy-25-00675-f002:**
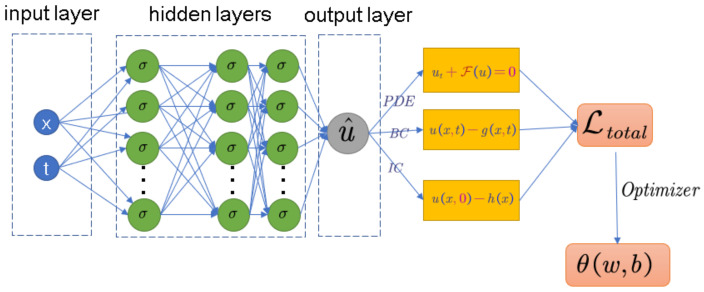
Physics-informed neural network structure diagram.

**Figure 3 entropy-25-00675-f003:**
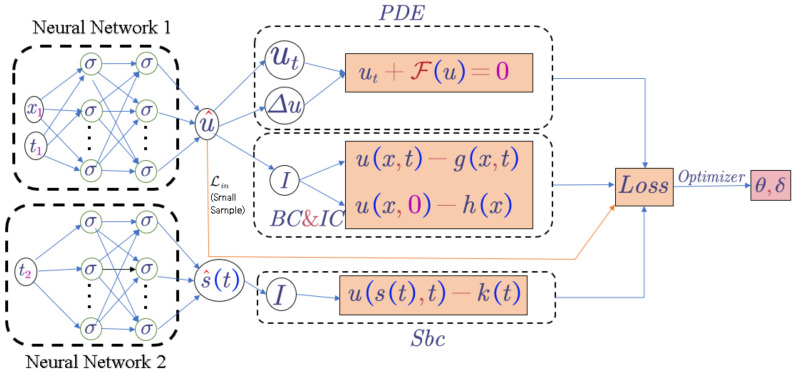
Stefan neural network structure diagram.

**Figure 4 entropy-25-00675-f004:**
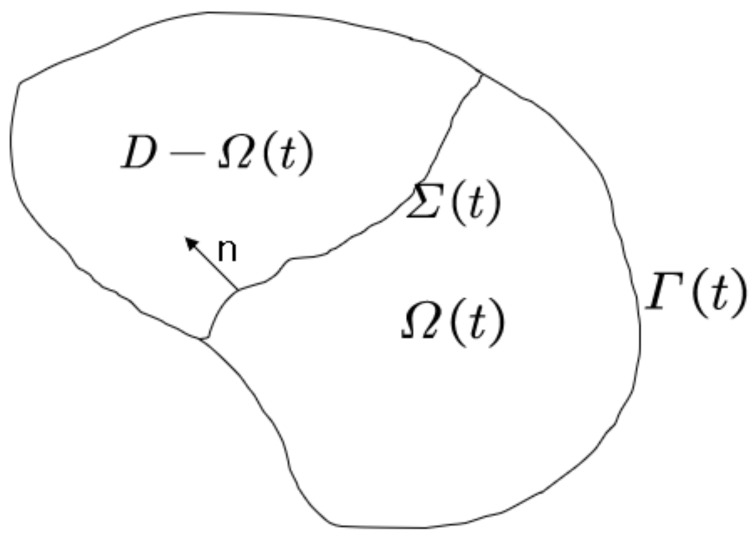
Stefan neural networks structure diagram.

**Figure 5 entropy-25-00675-f005:**
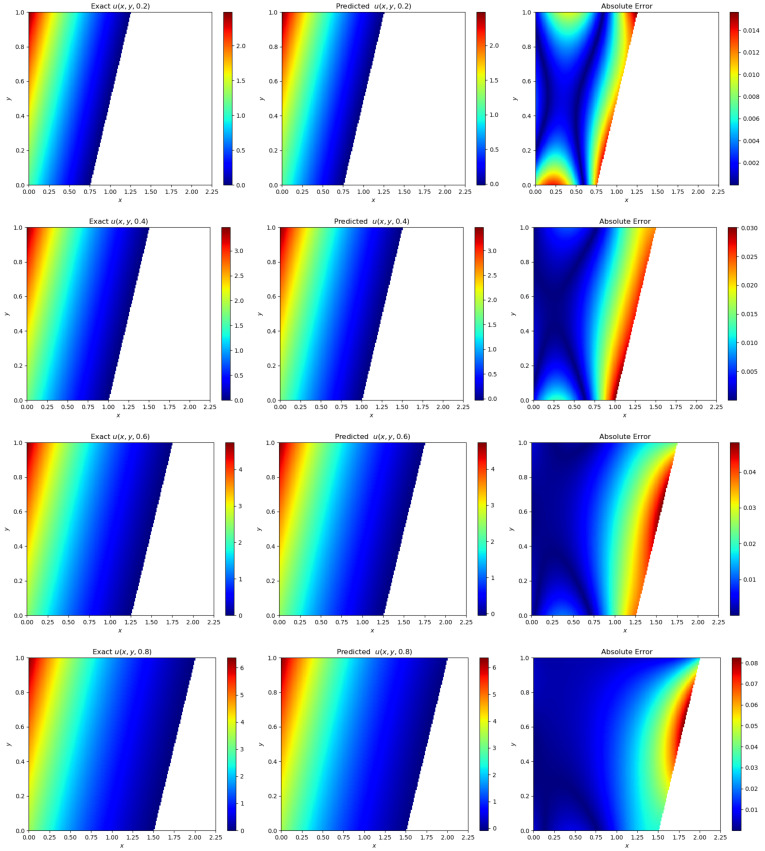
Improved physics-informed neural networks for solving the two-dimensional forward Stefan problem solution error diagram.

**Figure 6 entropy-25-00675-f006:**
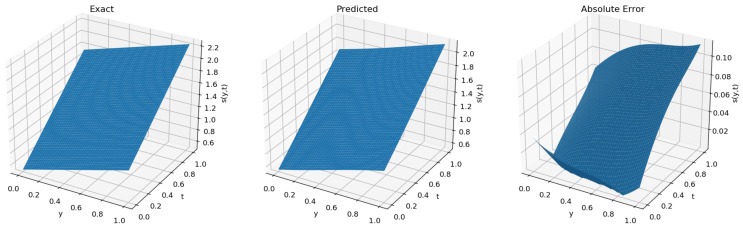
Improved physics-informed neural networks for solving the two-dimensional forward Stefan problem to predict the free boundary error diagram.

**Figure 7 entropy-25-00675-f007:**
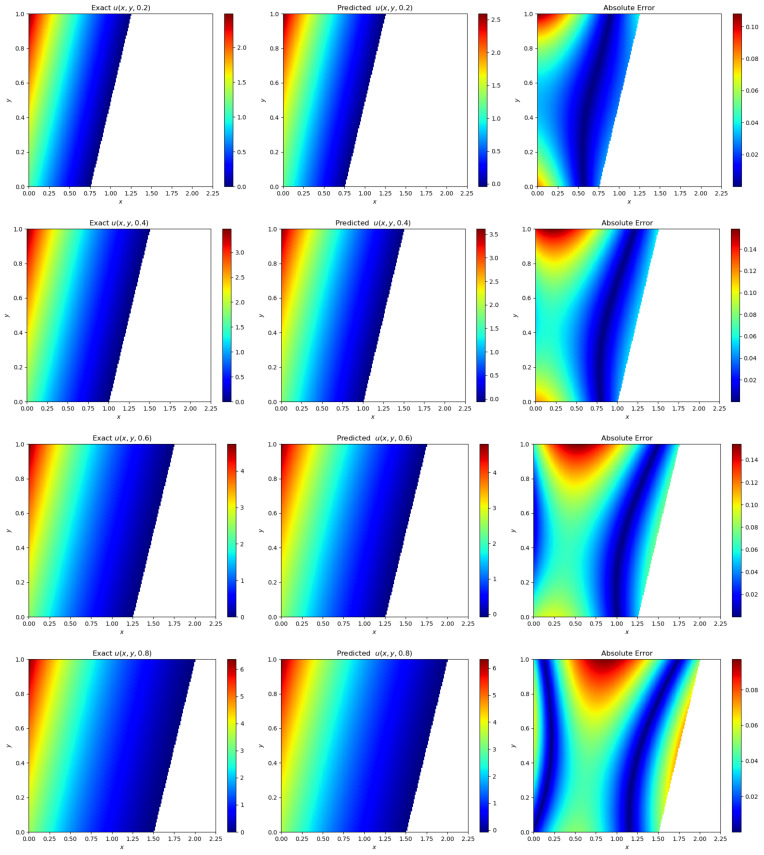
Improved physics-informed neural networks for solving the two-dimensional inverse Stefan problem solution error diagram.

**Figure 8 entropy-25-00675-f008:**
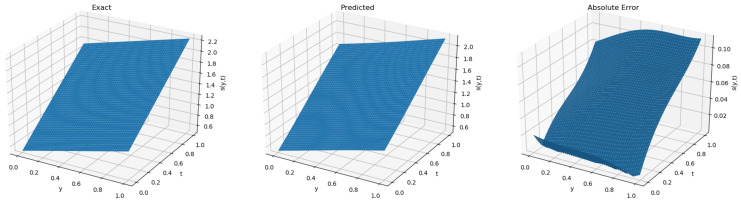
Improved physics-informed neural networks for solving the two-dimensional inverse Stefan problem I to predict the free boundary error diagram.

**Figure 9 entropy-25-00675-f009:**
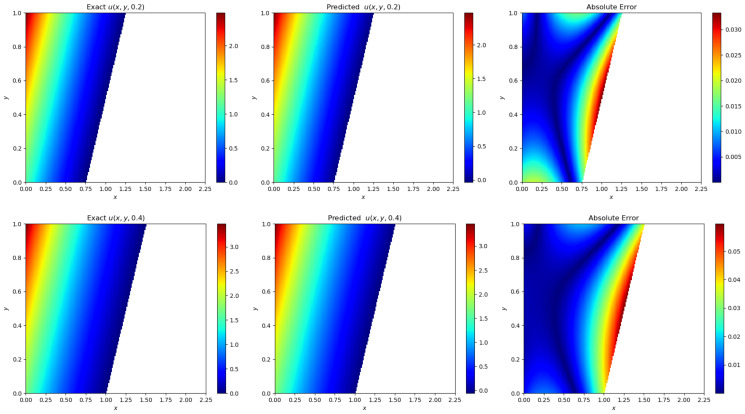
Improved physics-informed neural networks for solving the two-dimensional inverse Stefan problem solution error diagram.

**Figure 10 entropy-25-00675-f010:**
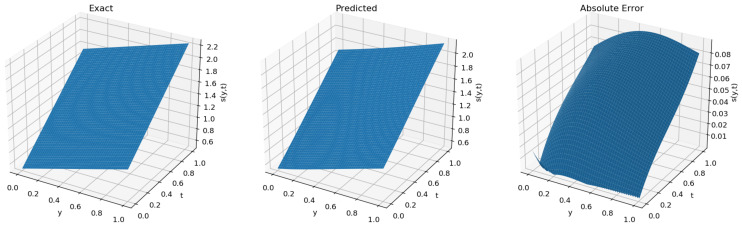
Improved physics-informed neural networks for solving the two-dimensional inverse Stefan problem II to predict the free boundary error diagram.

**Figure 11 entropy-25-00675-f011:**
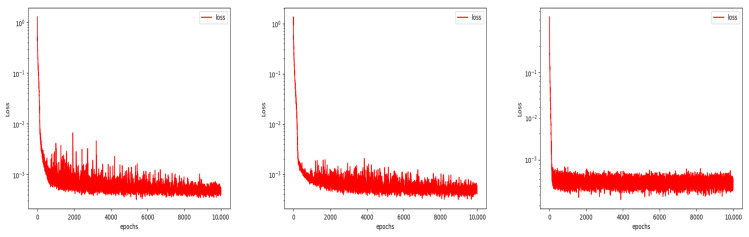
The two-dimensional inverse Stefan problem II with a 10% noise level of the MSE, log-cosh and log-cosh loss function with the addition of small sample loss decreases the plot.

**Figure 12 entropy-25-00675-f012:**
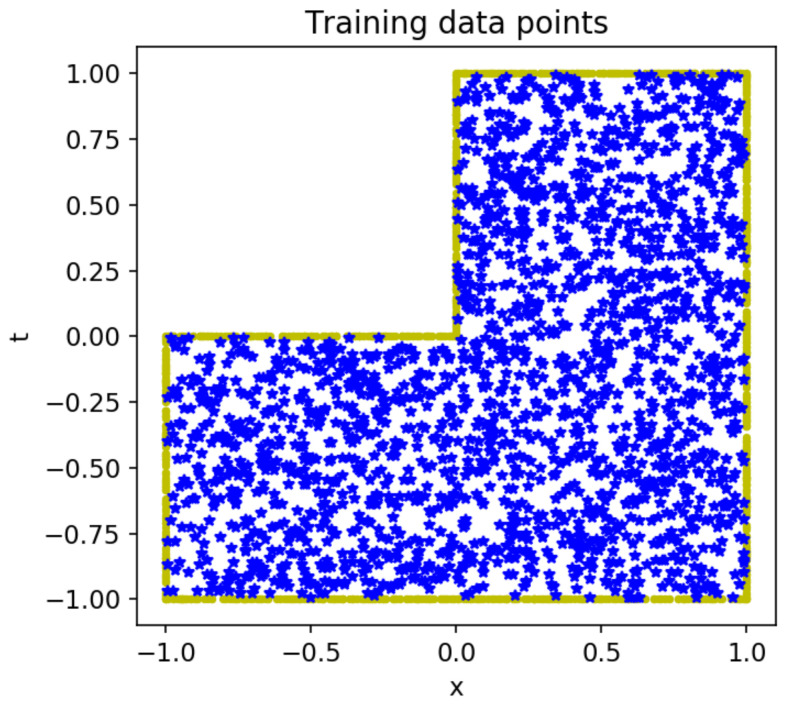
Red circled points indicate boundary points and initial points. Blue star-shaped points indicate internal points.

**Figure 13 entropy-25-00675-f013:**
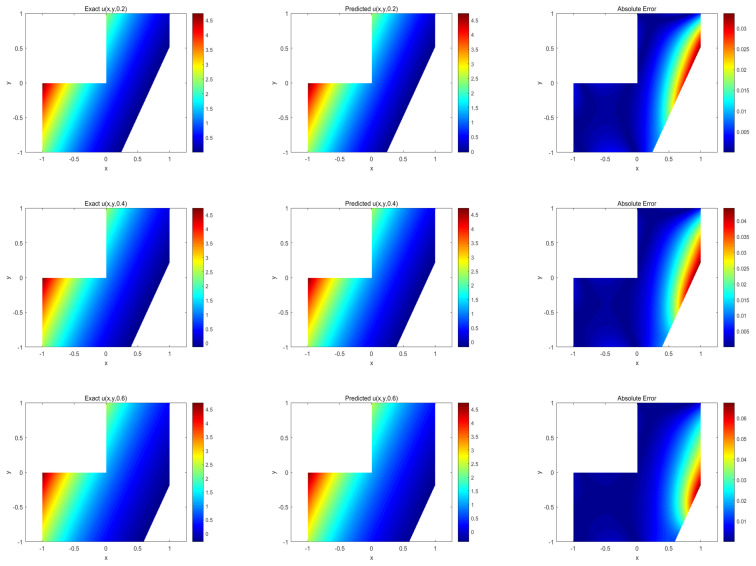
Exact solution, prediction solution, point-by-point error, and free boundary location for different moments of the temperature in the irregular region of the Stefan problem.

**Figure 14 entropy-25-00675-f014:**
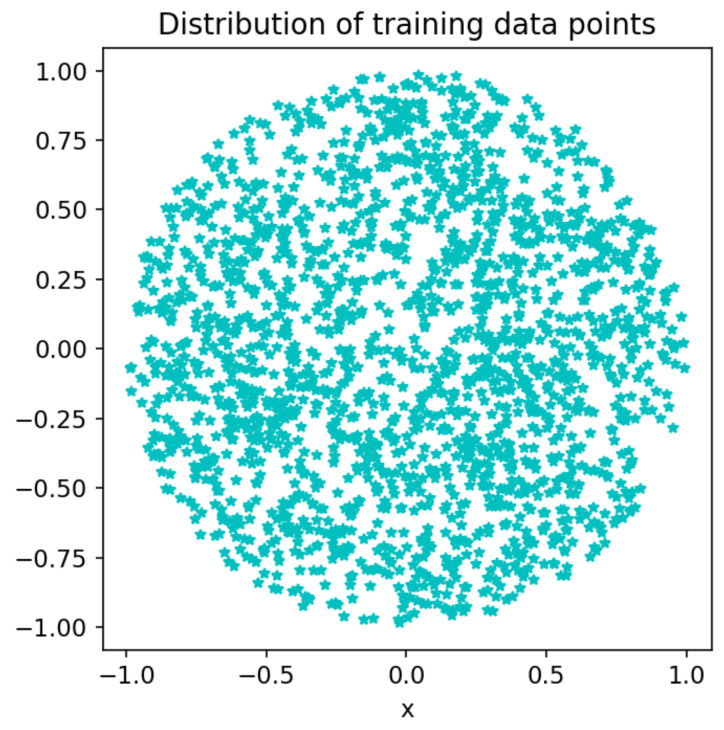
Scatter plot of training points.

**Figure 15 entropy-25-00675-f015:**
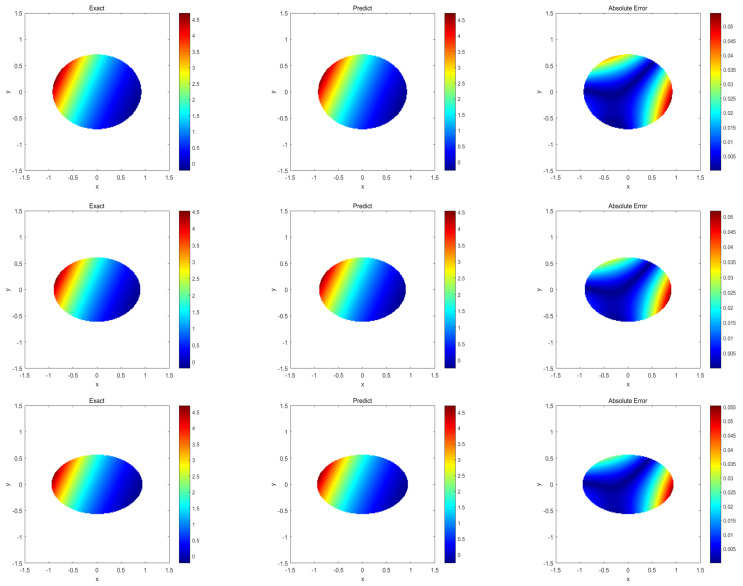
Exact solution, prediction solution, point-by-point error, and free boundary location for different moments of temperature in the irregular boundary of the Stefan problem.

**Table 1 entropy-25-00675-t001:** Effect of different small sample data points on two-dimensional forward Stefan problem.

Small Sample Data	0 (Original [34])	50	100	150	200
Solutions’ $L^{2}$ error	$8.12\times 10^{-2}$	$1.12\times 10^{-2}$	$8.57\times 10^{-3}$	$9.87\times 10^{-3}$	$3.32\times 10^{-2}$
Boundary $L^{2}$ error	$4.32\times 10^{-2}$	$4.22\times 10^{-2}$	$3.25\times 10^{-2}$	$4.33\times 10^{-2}$	$4.45\times 10^{-2}$

**Table 2 entropy-25-00675-t002:** Effect of different small sample data points on the two-dimensional inverse Stefan problem I.

Small Sample Data	0 (Original [34])	50	100	150	200
Solutions’ $L^{2}$ error	$8.89\times 10^{-2}$	$3.82\times 10^{-2}$	$3.66\times 10^{-2}$	$4.49\times 10^{-2}$	$4.59\times 10^{-2}$
Boundary $L^{2}$ error	$3.41\times 10^{-2}$	$3.35\times 10^{-2}$	$3.15\times 10^{-2}$	$3.32\times 10^{-2}$	$3.36\times 10^{-2}$

**Table 3 entropy-25-00675-t003:** Effect of different small sample data points on the two-dimensional inverse Stefan problem II.

Small Sample Data	0 (Original [34])	50	100	150	200
Solutions’ $L^{2}$ error	$6.12\times 10^{-2}$	$5.12\times 10^{-2}$	$9.97\times 10^{-3}$	$4.32\times 10^{-2}$	$7.62\times 10^{-2}$
Boundary $L^{2}$ error	$3.04\times 10^{-2}$	$2.13\times 10^{-2}$	$1.26\times 10^{-2}$	$2.56\times 10^{-2}$	$4.25\times 10^{-2}$

**Table 4 entropy-25-00675-t004:** Inverse problem II relative $L^{2}$ error of the prediction solution $u\left(x,t\right)$ for different measurement values *M* and different noise levels κ.

	κ	κ=0%	κ=1%	κ=2%	κ=5%	κ=10%
*M*						
M=50		$9.97\times 10^{-3}$	$8.75\times 10^{-2}$	$9.87\times 10^{-2}$	$1.95\times 10^{-1}$	$5.23\times 10^{-1}$
M=100		$2.64\times 10^{-2}$	$8.87\times 10^{-2}$	$9.98\times 10^{-2}$	$1.51\times 10^{-1}$	$8.36\times 10^{-1}$
M=200		$1.85\times 10^{-2}$	$2.26\times 10^{-2}$	$4.21\times 10^{-2}$	$5.75\times 10^{-2}$	$6.78\times 10^{-2}$

**Table 5 entropy-25-00675-t005:** Inverse problem II relative $L^{2}$ error of the free boundary $\alpha \left(t\right)$ for different measurement values *M* and different noise levels κ.

	κ	κ=0%	κ=1%	κ=2%	κ=5%	κ=10%
*M*						
M=50		$1.26\times 10^{-2}$	$1.43\times 10^{-2}$	$4.28\times 10^{-2}$	$1.72\times 10^{-1}$	$1.9\times 10^{-1}$
M=100		$1.48\times 10^{-2}$	$3.87\times 10^{-2}$	$5.11\times 10^{-2}$	$8.41\times 10^{-2}$	$1.21\times 10^{-1}$
M=200		$1.23\times 10^{-2}$	$1.35\times 10^{-2}$	$2.72\times 10^{-2}$	$2.81\times 10^{-2}$	$3.77\times 10^{-2}$

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
