# Peer review of "Improved Physics-Informed Neural Networks Combined with Small Sample Learning to Solve Two-Dimensional Stefan Problem"

_entropy, 2023, doi:10.3390/e25040675_

Round 1

Reviewer 1 Report

This paper proposes a neural network framework to solve the Stefan problem by improving the structure of physics-informed neural networks (PINNs) and combining small-sample learning with deep learning. The experiments show that this approach improves computational accuracy by 2-3 times and enhances model generalization ability.

Strengths:

1. Novel usage of PINN in a specific domain.

2. Interesting settings with small data

Weakness:

I'm generally satisfied with the method, however, more experiments on the added ssl loss should be justified, possibly with the sensitivity on \lambda_{ssl} and more direct descriptions in Section 3.3.2.

Reviewer 2 Report

The authors propose physics-informed neural networks, joining small-sample learning with a deep learning method, to solve the Stefan problem.

Numerical experiments illustrate that the improved deep neural networks based on small sample learning improves the computational accuracy by a factor superior to 2 compared with the original neural networks. The performance on high-dimensional problems and special irregular moving interface problems remains to be verified.

The proposed approach is highly inspired and based on [Raissi, M.; Perdikaris, P.; Karniadakis, G.E.; Physics-informed neural networks: a deep learning framework for solving forward and inverse problems involving nonlinear partial differential equations. J. Comput. Phys. 2019, 378, 686–707] and therefore the contribution proposed by the authors should be clearly stated. This is crucial both to highlight the contribution and to distinguish from the cited reference. Furthermore, the sentences used by the authors to supposedly answer this observation are unreadable (lines 88 to 97).

As much as I can understand, the main difference is the consideration of a loss function with small sample data towards an approach – “small-sample learning with a deep learning method small-sample learning”. With this approach the authors illustrate numerically that their strategy can be advantageous, which can be expected to be true for small sample data points. However, the performance of the proposed approach facing problems high-dimensional problems or irregular moving interface problems must be assessed. Without this more broader approach, the results are not sufficiently relevant.

Minor comments:

Page 1, line 2: consider “deep neural networks” instead of “a deep neural networks”

Page 1, lines 28 to 30, 37 o 38: consider the expected space before the citations

Page 2, line 46: remove the space before temperature in “( temperature ”

Page 2, line 61: consider the expected space before citation [28]; also for line 84.

Page 2, line 74: consider “become popular [29–32]. Raissi “ instead of “become popular[29–32], Raissi ”

Page 2, line 88 and beyond: The sentence is crucial to the paper since it states the novelty proposed. Yet, it is unreadable: “In this paper, we propose a physics-informed neural networks that combines small sample learning and is different from PINNs [33] for solving the Stefan problem, incorporating the ideas of PINNs [33] for solving partial differential equations into the neural networks framework for solving the Stefan problem”. Maybe the authors would like to write something similar to “In this paper, we propose a physics-informed neural network that combines small sample learning with a deep learning method, following closely the ideas from PINNs [33] for solving the Stefan problem.”

Page 3, lines 117 to 121: “Each Stefan problem corresponds to a class of inverse problems, and similar to other mathematical physics inverse problems, the inverse Stefan problem is not definite, and the uniqueness and stability of the solution are not always guaranteed. The uniqueness and stability of the solution are not always guaranteed to hold.” Page sentences repeated.

Page 5, line 155: Consider a sentence break; “… moving boundary a(t). To this end”

Page 6, line 174: duplication: “while not being affected by too many outliers, without being affected by too many outliers.”

Page 9, line 239: consider spaces “Figure 5 and 6,”

Round 2

Reviewer 2 Report

The authors made a correction effort based on the comments provided. In particular, they clarify the objective of the work and its contribution. The numerical results were greatly enriched.